# Systemic Inflammation Predicts Alzheimer Pathology in Community Samples without Dementia

**DOI:** 10.3390/biomedicines10061240

**Published:** 2022-05-26

**Authors:** Nicolas Cherbuin, Erin I. Walsh, Liana Leach, Anne Brüstle, Richard Burns, Kaarin J. Anstey, Perminder S. Sachdev, Bernhard T. Baune

**Affiliations:** 1Research School of Population Health, Australian National University, Canberra, ACT 2601, Australia; erin.walsh@anu.edu.au (E.I.W.); liana.leach@anu.edu.au (L.L.); richard.burns@anu.edu.au (R.B.); 2John Curtin School of Medical Research, Australian National University, Canberra, ACT 2601, Australia; anne.bruestle@anu.edu.au; 3School of Psychology, University of New South Wales, Sydney, NSW 2052, Australia; k.anstey@unsw.edu.au; 4Neuroscience Research Australia, Sydney, NSW 2052, Australia; 5Centre for Healthy Brain Ageing (CHeBA), Discipline of Psychiatry & Mental Health, University of New South Wales, Sydney, NSW 2052, Australia; p.sachdev@unsw.edu.au; 6Department of Psychiatry, University of Münster, 48149 Münster, Germany; bernhard.baune@ukmuenster.de; 7Department of Psychiatry, Melbourne Medical School, The University of Melbourne, Melbourne, VIC 3010, Australia; 8The Florey Institute of Neuroscience and Mental Health, The University of Melbourne, Melbourne, VIC 3052, Australia

**Keywords:** inflammation, oxidative stress, Amyloid beta, total tau, immunoassays, middle-age, plasma

## Abstract

Neuroinflammation and oxidative stress (OS) are implicated in the pathophysiology of Alzheimer’s disease (AD). However, it is unclear at what stage of the disease process inflammation first becomes manifest. The aim of this study was to investigate the associations between specific plasma markers of inflammation and OS, tau, and Amyloid-β 38, 40, and 42 levels in cognitively unimpaired middle-age and older individuals. Associations between inflammatory states identified through principal component analysis and AD biomarkers were investigated in middle-age (52–56 years, *n* = 335, 52% female) and older-age (72–76 years, *n* = 351, 46% female) participants without dementia. In middle-age, a component reflecting variation in OS was most strongly associated with tau and to a lesser extent amyloid-β levels. In older-age, a similar component to that observed in middle-age was only associated with tau, while another component reflecting heightened inflammation independent of OS, was associated with all AD biomarkers. In middle and older-age, inflammation and OS states are associated with plasma AD biomarkers.

## 1. Introduction

Dementia represents a major burden of disease and is associated with huge personal, social, and economic costs. Worldwide, almost 44 million [1] people live with dementia and this figure is forecast to increase to 150 million by 2050 [2]. Dementia is the fifth leading cause of death and its annual cost to society has been estimated at $818 million globally [1].

Alzheimer’s disease (AD) is the most prevalent type of dementia and accounts for 60–70% of all cases. Neuropathologically, it is characterized by the presence of two hallmarks, the extracellular deposition of Amyloid beta (Aβ) plaques in brain tissue, and the formation of neurofibrillary tangles inside neurons as a product of tau hyperphosphoralisation [3]. Aβ is a monomer of varying length ranging from 37 to 49 amino acids, but most frequently 40 or 42. It contributes to lipid transport and normal metabolism under physiological conditions. However, it can accumulate in soluble form or in aggregate when the main mechanisms regulating its concentration, proteolysis and clearance into the cerebro-spinal fluid and the bloodstream start failing [4]. The early amyloid hypothesis suggested that Aβ plaques were toxic, and thus led to neurodegeneration, but current views most strongly support a role of highly toxic soluble Aβ, and particularly its Aβ 42 variant [5].

In contrast, the tau protein presents six different isoforms dependent on their terminal structure in the brain, and they range in length between 352–441 amino acids. One of its functions, among others, is to bind transiently to axonal micro-tubules and to stabilise them [6]. Tau phosphoralisation occurs in mammals during some physiological processes including development, hibernation, and hypothermia [7]. However, there is accumulating evidence that tau hyperphosphoralisation is increased and prevalent in AD [7], which impairs tau binding to micro-tubules, and enables its combination into oligomers, which are thought to be most toxic, and its aggregation into filaments or fibrils [8].

While there is a large body of evidence pointing to both Aβ and tau being implicated in AD neuropathology, the extent to which they are causal to the disease or by-products of associated mechanisms is not completely resolved and the subject of heated debates. There is also substantial divergence of opinion as to whether Aβ or tau contribute most to the disease process [5,6,9,10,11]. However, it appears increasingly likely that an interaction between Aβ and tau, following their spread and progressive spatial confluence after developing in somewhat different brain regions, accelerates neurodegenerative processes [12]. Further complicating this complex picture is the fact that AD is not a homogenous syndrome with cardio-metabolic and cerebro-vascular disease making substantial contributions to the disease process. Indeed, the vast majority (>80%) of individuals with AD pathology present with co-morbid cardio-vascular disease (CVD) [13], and cardio-vascular and metabolic dysfunctions have been confirmed as major early to mid-life dementia risk factors [14].

Robust evidence is available showing that Aβ, tau, and cerebrovascular disease are individually and in combination associated with microscopic neuronal changes (e.g., shrinking dendritic tree, decreasing synaptic density, and demylenation), as well as microscopic structural brain changes demonstrated in histological and neuroimaging studies (e.g., total and regional brain volume shrinkage, decrease structural connectivity, widening sulcal width, and increased ventricular and cerebrospinal volume [3,15,16]. Importantly, these brain changes start occurring well before the development of clinical AD in individuals with normal cognition [17,18]. Moreover, the type of brain changes associated with Aβ, tau, and cerebrovascular disease are predictive of cognitive decline and of progression to mild cognitive impairment (MCI) and AD [18,19,20,21,22].

Central to the focus of the present study, neuroinflammation has been shown to contribute to AD and other dementias’ pathophysiology [23,24]. There is also developing evidence that systemic inflammation is a risk factor for neurodegeneration and cognitive impairment, and is closely associated with major modifiable risk factors for dementia (e.g., CVD, type 2 diabetes, obesity, and depression) [25,26]. However, the extent to which inflammation is attributable to disease progression in the central nervous system (CNS) in the pre- and clinical stages of dementia or emerges in the periphery and is a long-term risk factor for the development of AD pathology is an unresolved question [27,28].

Aβ and phosphorylated tau up-regulate inflammatory mechanisms in the CNS. Aβ, whether in soluble or aggregated form, activates microglia, which leads to prolonged inflammation and increased oxidative stress (OS), and ultimately promotes apoptosis and neurodegeneration [27,29]. Hyperphosphorylated tau aggregates are also able to activate micro-glia when they are released from dead neurons [30], and conversely inflammation increases tau phosphorylation [31]. Similarly, hyperphosphorylated tau appears to increase oxidative stress levels, while OS can also promotes tau hyperphosphylisation and aggregation [8]. In addition, cerebral small vessel disease, a known dementia risk factor [32] that co-occurs with amyloid plaques and tau tangles [33], is also associated with heightened inflammation. While the exact nature of this relationship has not been fully elucidated, it is likely that small vessel disease is both a promoter and a consequence of neuroinflammation [34]. Thus, CNS inflammation, OS, and AD pathology appear to be clearly related mechanistically and to be involved in feedback loops that mutually accelerate their production.

The precise temporal and spatial relationship between these processes is not well understood, but emerging evidence suggests that inflammation may precede Aβ deposition and may be caused or exacerbated by increased tau phosphoralisation. Indeed, a recent study [35] investigating the associations between central inflammation with ^11^C-PK11195 Positron Emission Tomography (PET), Aβ (^11^C-PiB PET), and tau (^18^F-Flortaucipir PET) deposition in MCI over a two-year period indicates that neuroinflammation may occur in the early AD pre-clinical stages, and is followed by a spatially correlated increase in Aβ in the frontal, temporal, and occipital lobes. Interestingly, these associations do not seem to persist when controlling for tau load. In contrast, central inflammation was found to be spatially correlated with tau deposition across the whole follow-up, first in the frontal and temporal lobes, and later in the occipital lobe. However, unlike for Aβ, associations between central inflammation and tau did not change after controlling for Aβ load. This appears to suggest that inflammation may be more strongly associated with tau pathology than Aβ. Evidence from bio-chemistry, in-vitro, animal, and other human studies also seem to bring support to this hypothesis [36], and consistent evidence has also been demonstrated in fronto-temporal dementia [24].

In addition, there is accumulating evidence indicating that systemic inflammation also contributes to, or at least co-occurs with AD pathophysiology [37,38]. In animal models, systemic inflammation has been shown to be associated with glial activation, tau hyperphosphoralisation in the frontal cortex and the hippocampus, and Aβ deposition in the hippocampus [39,40], while human studies have reported associations between markers of systemic inflammation and PET markers of Aβ and tau deposition [41]. Moreover, blood cytokines are small molecules known to cross the blood-brain barrier [42], and a recent meta-analysis of 175 studies including 13,344 participants with AD and 12,912 healthy controls demonstrated that several blood cytokines and chemokines were consistently elevated in AD, and that higher inflammation levels were associated with lower mini-mental examination [26]. However, inconsistent associations have been detected in mild cognitive impairment [43] (MCI) and it is currently unclear whether these findings should be interpreted as showing that systemic inflammation is not implicated at the MCI stage or whether these results reflect the high degree of inhomogeneity observed in MCI studies. Studies in animals and in at risk human populations, e.g., those with type 2 diabetes, obesity or hypertension, which demonstrate increased development of AD pathology in the context of peripheral inflammation, would suggest that the latter is more likely. Similarly, while some inflammatory markers (e.g., interleukin 6, IL6) appear to be associated with decline in cognitive function in community-living individuals [44], inconsistent results are also prevalent in this population [45].

Therefore, to clarify whether systemic inflammation is implicated in the early development of AD pathology there is a need to investigate this question in large, detailed investigations of cognitively unimpaired individuals living in the community. A particular limitation of current research in this area is that available studies have often either focused on very few and frequently general markers of systemic inflammation (e.g., C-reactive protein, CRP), or on multiple specialised inflammatory markers (e.g., cytokines) tested individually [26]. Both approaches are problematic because they are either not specific enough, or because they do not appropriately reflect the fact that cytokines and chemokines are neither good nor bad in themselves, but interact together to produce inflammatory states that may be protective or deleterious [46].

In a previous study [25], we have shown that a large number of markers could be used to identify, through principal component analysis, different inflammatory states. We were then able to show that these states were differentially associated with MCI, and longitudinal decline in MMSE and hippocampal volume. Importantly, this was achieved while also appropriately controlling for multiple comparisons and recognising the complex inter-relationship between biomarkers. To further add to this evidence, the aim of this study is to determine whether systemic inflammation is associated with the development of AD pathology, as indicated by plasma tau and Aβ38, 40 and 42 levels, in samples of midlife and older individuals without neurological disorders by applying a principal component analysis of a large number of inflammatory markers. In this context it is important to note that while tau and Aβ40 and 42 proteins are also produced outside the CNS, they mostly originate from the brain, and plasma levels have been shown to reflect both cerebrospinal levels [47] and brain load as assessed by proton-emission tomography (PET) [48,49,50], and to significantly differ in the pre-clinical AD stages, although less consistently in the more advanced stages of the disease [51,52,53].

## 2. Materials and Methods

### 2.1. Study Population

Participants included in the present study were selected from the larger PATH Through Life (PATH) project, which has been described elsewhere [54]. Briefly, PATH randomly sampled individuals from the electoral roll of the city of Canberra and adjoining town of Queanbeyan across three age groups. The focus of this investigation is on samples of middle-age (MA: *n* = 2530; 53–56 years) and older-age (OA: *n* = 2550; 73–76 years) participants drawn from two population-based cohorts for whom detailed pro-inflammatory, AD biomarkers, and covariates were available (MA: *n* = 230; OA: *n* = 409) at the fourth wave of data collection. Participants were excluded if they had neurological conditions (stroke, MMSE < 25, either Parkinson’s or Dementia diagnosis at any point in the study (MA: *n* = 0, OA *n* = 29). This resulted in a final sample of 610 participants (MA: *n* = 230, 54% female; OA *n* = 380, 44% female). Compared with the broader PATH cohorts at baseline MA, selected MA participants did not differ from their non-selected counterparts in age, sex or education. Selected OA participants were slightly younger (~four months, t = 3.81, *p* < 0.001) but did not differ in sex or education.

### 2.2. Inflammatory Cytokines, OS, and AD Biomarkers

Markers of inflammation: Tumour Necrosis Factor alpha (TNF-α) and TNF receptors (TNF-R1, TNF-R2), and interleukins (IL1β, IL4, IL6, IL8, IL10). OS markers: nitric oxide (NO), neopterin (NEO), total anti-oxidant capacity (TAC). OS-related DNA damage: malondialdehyde (MDA), 8-hydroxy-2-deoxyguanosine (GUA). AD peptides: Aβ38, 40, 42 and 40/42 ratio, as well as total tau. All biomarkers were assessed in duplicates using highly sensitive validated immunoassays designed to detect constituents in the picogram (pg) range on the Mesoscale platform (V-PLEX; Rockville, MD, USA) as recommended by the assay manufacturer and previously described [25]. Serum/plasma samples were collected after a fast of at least eight hours and stored at −80 °C aliquoted in 1 mL vials. Immediately prior to analysis, samples were thawed and were all processed with the same pipeline at the same time at the ANU Phenomics Facility (see Appendix A for details). CRP was also measured as a non-specific marker of systemic inflammation.

### 2.3. Socio-Demographic and Health Measures

Age, total years of education, diabetes mellitus, depression symptomatology (Goldberg depression) [55], and smoking (ever) were assessed by self-report. Body mass index (BMI) was computed with the formula weight (kg)/height × height (m^2^) based on self-report of weight and height.

### 2.4. Identification of Pro-Inflammatory States

An identical approach was used to identify pro-inflammatory states as that previously published on the OA sample of the PATH study [25]. Briefly, the principal components of the observed cytokine response across all inflammatory and oxidative stress markers were extracted through a principal component analysis (PCA) using the R package “prcomp” [56]. Principal components were selected based on an eigen value > 1 (see Appendix A).

### 2.5. Statistical Analysis

Statistical analyses were computed using the R statistical package (version 4.1.2, R Core Team, Vienna, Austria) under Rstudio (version 2021.09.1.372, RStudio Team, Boston, MA, USA). Missing values (<1% of all variables analysed) were imputed by chained equations with the package “mice” using the “pmm” algorithm [57]. Descriptive analyses were conducted using Chi-square tests for categorical data and t-tests to compare groups on continuous variables. Associations between principal components (inflammatory states) and outcome measures (tau, Aβ38, 40, 42 and 40/42 ratio) were investigated with linear regression analyses while controlling for covariates assessed at the same time as the biomarkers including age, sex, education in an initial model, and in addition controlling for diabetes, body mass index (BMI), smoking, depression, alcohol intake, and physical activity in a fully adjusted model. Sensitivity analyses were conducted to assess associations between CRP and inflammatory states identified through PCA. Additional information is provided in the Appendix A. Alpha was set at *p* < 0.05 and corrected for multiple comparisons (Bonferroni).

## 3. Results

Participants’ demographic measures are presented in Table 1. MA had, on average, a higher education level than OA, but had a lower BMI, and were less likely to be hypertensive or to have diabetes. Across all participants, men had a higher education level, a higher BMI, and were more likely to be hypertensive than women.

### 3.1. Principal Component Analysis

Pearson bivariate correlations between the inflammatory markers are presented in Appendix A.

Results of the PCA analysis are presented in Appendix A. Using an eigen-value cut-off of 1, four main principal components (PC1–PC4) reflecting different inflammatory states were identified which explained 60.63% of the variance in pro-inflammatory and OS markers in the 40s, and 54.86% in the 60s. The relative contribution of the different markers to each PC is presented in Appendix A. While some variation was observed, the four principal components were comparable between age samples (see Appendix A for a detailed discussion). PC1 can be interpreted as consisting of relatively unspecific heightened pro-inflammatory response, particularly involving TNFα and TNF receptors, in the context of low OS activity. PC2 suggests a pro-inflammatory response associated with somewhat increased (MA) or decreased (OA) anti-oxidant activity. PC3 is indicative of increased OS and/or decreased antioxidant activity (NO/TAC) associated with DNA damage (MDA, GUA) in the context of a low inflammatory response. Furthermore, PC4 is indicative of higher anti-oxidant activity (TAC), lower OS (NO), and increased DNA damage (GUA/MDA). Bivariate Pearson correlations between the principal components and AD biomarkers are presented in Appendix A.

### 3.2. Inflammatory States & AD Biomarkers in Middle Age (MA)

Analyses testing associations between the main PCs and AD biomarkers are presented in Table 2. PC1 was significantly associated with Aβ40; PC3 was significantly associated with Aβ40 & 42; and PC4 was significantly associated with tau, Aβ38, 40 & 42. A schematic of these associations and their direction is presented in Figure 1.

### 3.3. Inflammatory States & AD Biomarkers in Older Age (OA)

Analyses testing associations between the main PCs and AD biomarkers are presented in Table 3. PC1 was significantly associated with tau, Aβ38, 40 & 42; and PC4 was significantly associated with tau, Aβ38. A schematic of these associations and their direction is presented in Figure 1.

### 3.4. C-Reactive Protein & Inflammatory States

Additional analyses were conducted to determine whether a non-specific marker of systemic inflammation (CRP), which is widely used in the clinic, was associated with the inflammatory states identified in the PCA. No associations were detected between CRP and any of the principal components (PC1-PC4; Appendix A) in MA or OA.

### 3.5. Sensitivity Analyses

As renal clearance differs between individuals, decreases with increasing age, and influences plasma AD biomarker levels [58,59], additional regression analyses were conducted to determine whether controlling for plasma creatinine levels influenced the relationship between PCs and AD biomarkers. These analyses produced essentially the same results as those presented in Table 2 and Table 3 except for minor decimal differences (results not shown).

## 4. Discussion

This study has applied a very novel approach to investigate how systemic inflammation relates to biomarkers of two AD hallmarks. By considering inflammatory states, identified through PCA and assessed through the combined contribution of several markers of inflammation and OS (i.e., not single cytokines individually), it ensured that the complex inter-relationship between cytokines and chemokines involved in varying inflammatory responses would be captured in a more holistic way. Another novel aspect of this research is that these relationships were investigated in community-living individuals without dementia.

Three main findings emerged from this study. First, four similar, complementary inflammatory states (principal components) were identified in middle-age and older-age participants. Secondly, some, but not all, of these inflammatory states were significantly associated with AD biomarkers with notable differences between age samples. And thirdly, a widely used marker of systemic inflammation (CRP) was not associated with any of the identified inflammatory states.

It is particularly notable that despite an age difference of 20 years between the two age samples, remarkably similar inflammatory states were detected among them. Discriminating qualities between the inflammatory states appear to relate to the degree to which they integrate one or more broad pro-inflammatory, OS, or OS-related DNA damage signals. Indeed, the first component identified involves almost exclusively a pro-inflammatory response (interleukin and TNF activation). In contrast, the second component, while also reflecting a broad pro-inflammatory response, involves anti-oxidant activity. A possible interpretation is that the part of the pro-inflammatory response contributing to this component is up-regulated by an increase in OS, which is not sufficiently buffered by the anti-oxidative response [60]. The third state is characterised by increased OS, decreased anti-oxidant activity, and increased DNA damage. It may be interpreted as the portion of the OS signaling that does not contribute to the upregulation of the inflammatory response in the context of insufficient anti-oxidant activity and consequently increased DNA damage. Finally, the fourth component also involves DNA damage, but in the context of low OS, and higher anti-oxidant activity, which may indicate OS-related DNA damage that is not attributable to current OS activity, possibly dampened by an increased anti-oxidant response.

Of even greater significance is that some of the identified inflammatory states were strong predictors of AD biomarkers. This is not completely surprising since we have shown previously [25], using the same approach, that these inflammatory states were related to cognitive impairment and neurodegeneration in the same older-age participants as those included in this study. Moreover, it is also consistent with the animal literature demonstrating that heightened inflammation is associated with increased amyloid and tau load in transgenic mice brains [61,62]. However, it is worth highlighting that associations between inflammatory states and AD biomarkers detected in the present analyses varied between the age samples. In the middle-age sample, the first component (unspecific heightened pro-inflammatory response) was only slightly associated with Aβ40 levels, while in the older-age sample it was moderately associated with all AD biomarkers. The reason for this difference is unclear but it is likely to relate to differences in brain and systemic clearance efficiency. Indeed, substantial age-related decreases in cerebrospinal fluid clearance have been reported in mice as well as in humans, which is known to lead to greater brain exposure to the toxic effects of Aβ [63,64]. Similarly, renal clearance decreases with ageing and is associated with an increase in Aβ levels [65]. These differences in clearance may lead to blood levels of AD biomarkers to be more sensitive indicators of pathological changes associated with systemic inflammation/OS.

In contrast to the first, the fourth component (OS-related DNA damage) was very strongly associated with tau and Aβ38, and to a lesser extent with Aβ40 and 42 in the middle-age sample, but only showed a strong association with tau in the older-age sample. This may suggest that that the OS-related DNA damage that occurs despite increased anti-oxidant activity is particularly reflective of neuronal damage. This explanation would be consistent with the known role OS plays in tau phosphorylation, DNA damage, and progressive loss of efficiency of DNA repair mechanisms [66,67]. It should, however, be noted that the fourth component only explained 7–8% of the total variance in inflammation/OS biomarkers.

Finally, the third component (increased OS and DNA damage with decreased anti-oxidant activity) was moderately associated with Aβ40 and 42, but only in the middle-age sample. Since what mainly differentiates this component from the third is the decreased anti-oxidant response, it may indicate that when anti-oxidant levels are insufficient, either because of low endogenous production or low dietary intake relative to OS levels, this may contribute to greater Aβ production. This is in line with prior research demonstrating consistent associations between OS and Aβ levels [37], and is also consistent with our understanding that OS is particularly involved in the early stages of AD pathophysiology [68,69]. In contrast and unlike the other components, the second component (increased inflammation associated with increased anti-oxidant activity) was not associated with any of the biomarkers in the two samples. The most likely reason for this lack of association is that the part of the inflammatory response represented in the second component, which may be buffered to a greater extent than in the first component by the concurrent increase in anti-oxidant activity, does not contribute to the AD pathological processes, or at least not to the same extent as other components. This is supported by a strong evidence-base indicating that high anti-oxidant levels are associated with lower Aβ levels and AD pathology more generally [70,71].

Some of these findings warrant close attention. First, it is notable that the more consistent and robust associations involved oxidative stress, or DNA damage related to oxidative stress (third and fourth components). The literature tends to emphasise the contribution of inflammation to neurodegeneration and cognitive decline while often not specifically considering or measuring the involvement of oxidative stress. These results suggest that oxidative stress may be a particularly strong contributor to early AD pathophysiology and requires more systematic investigation. Secondly, high DNA damage that co-occurs with high anti-oxidant activity but lower oxidative stress (fourth component) seems to be most strongly related to tau levels, and less so with Aβ levels. In this context it is worth highlighting that this component was strongly associated with having mild cognitive impairment in our previous investigation [25]. Thirdly, it appears that increased inflammation associated with increased anti-oxidant activity (second component) is not associated with AD biomarker levels, whereas increased inflammation unrelated to oxidative stress (first component) is associated with increased AD biomarker levels, particularly in the older sample. This pattern could be expected, since inflammation that is buffered by anti-oxidant activity would be less likely to accelerate neurodegenerative processes. Nonetheless, it deserves highlighting since this evidence may bring further support to interventions aimed at up-regulating anti-oxidant pathways either through increased dietary intake, exercise, or pharmacological treatment. It is also interesting to note that non-specific inflammation (first component) was associated with hippocampal atrophy in our previous investigation using the same methodology in the same older participants [25].

A surprising finding is that the very widely used measure of systemic inflammation, C Reactive Protein, was not significantly associated with any of the inflammatory states identified, despite the fact it was assayed at the same blood draw as the other inflammatory markers. CRP is unspecific, and therefore low associations might have been expected. However, the complete lack of association suggests that it is not a marker suitable to assess systemically inflammatory states that may be associated with neurodegeneration and cognitive decline. An unresolved question is the extent to which the inflammatory states identified in the periphery in this investigation are the product of pathological processes taking place in the central nervous system or, alternatively, whether they are the origin of greater CNS inflammation.

### 4.1. Implications and Future Directions

A direct implication of the present findings is that, where possible, assessment of inflammation through multiple cytokines and chemokines is preferable to the use of single markers, or to the separate analysis of individual biomarkers. These results also raise important conceptual questions. Since cross-sectional analyses were conducted to assess associations between inflammatory states and AD biomarkers, it is not clear whether systemic inflammation is a consequence of the developing AD pathology, or whether an increase in peripheral inflammation led to, or promoted the development of AD pathology, or indeed, whether these processes are directly or causally related at all. Although there is theoretical support for all these hypotheses and they do not need to be mutually exclusive, based on current evidence it seems more likely that inflammation originating in the periphery has an effect on central pathological processes rather than the reverse. This is because there is robust evidence indicating that conditions associated with heightened peripheral inflammation, such as sepsis, osteo-arthritis, and systemic lupus erythematosus, are associated with increased neurodegeneration, cognitive decline and dementia [72,73,74,75]. Moreover, controlled experiments in AD mice models have shown that low-grade systemic inflammation is associated with greater glial activation, decreased Aβ clearance, and greater Aβ deposition [38,40]. In contrast, there is little, if any, evidence demonstrating that increased AD pathology may lead to increased systemic inflammation.

Nonetheless, there is a need for human studies investigating longitudinally the associations between systemic inflammation, increased AD pathology in the brain, and plasma AD biomarkers over longer timeframes and with several repeated assessments such that directional associations can be tested. Another research gap that requires more attention is the specific role OS plays in AD pathophysiology. OS is often conflated with inflammation and the present findings suggest that more routinely identifying the discrete effects attributable to each of these processes may prompt new insights into the timing and the origin of AD pathological development, and thus provide a better understanding of the optimal preventative window and of the more promising risk reduction strategies and targets.

### 4.2. Limitations

This study has a number of strengths and limitations. It investigated large samples of individuals drawn from the population whose age covered both middle- and older-age. However, the participants were predominantly Caucasian and highly educated, and therefore these findings may not be generalizable to other populations. Moreover, the cross-sectional design and the correlational nature of this study preclude any conclusion on causation or directionality. While there is good evidence indicating that plasma measures of tau and Aβ are indicative of CNS levels and corresponding AD pathologies, they do not exclusively reflect central production of these proteins and therefore may also reflect processes unrelated to AD pathology. Finally, although a large number of markers were considered, they represent only a fraction of all the components of the inflammatory response, and future research should aim to consider a broader set of biomarkers.

## 5. Conclusions

These findings provide further evidence that systemic inflammation and OS are related to AD biomarkers levels, which is also consistent with previous evidence indicating that heightened inflammation is associated with greater neurodegeneration and cognitive decline. Moreover, this research demonstrates that assessing holistically the combined contribution of different components of the inflammatory/OS response can improve our understanding of the mechanistic components (innate immunity, oxidative stress, DNA damage) that relate most strongly to neurodegenerative processes.

## Figures and Tables

**Figure 1 biomedicines-10-01240-f001:**
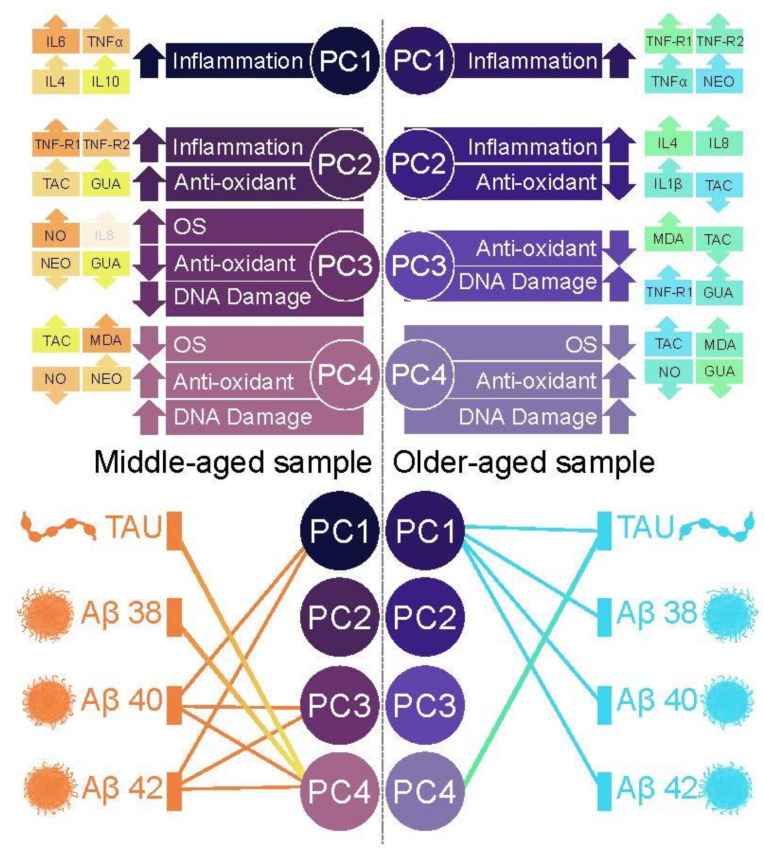
Top: Illustration of the identified inflammatory states (PC1–PC4) and their main contributing blood markers (inflammation: TNF-α, TNF-R1, TNF-R2, IL1-β, IL4, IL6, IL8, IL10; Oxidative Stress: NO, NEO; Anti-oxidant: TAC; DNA damage: GUA, MDA) in middle-age (**left**) and older-age (**right**) participants. Bottom: Significant associations (Bonferroni-corrected) between the identified inflammatory states (PC1–PC4) and AD biomarkers in middle-age (**left**) and older-age (**right**) participants.

**Table 1 biomedicines-10-01240-t001:** Participants’ characteristics.

Measures	40s (*n* = 230)	60s (*n* = 380)	T/chi-sq Test (*p* Value)
Age, years (SD)	55.77 (1.39)	75.35 (1.41)	−167.64 (0.000)
Education, years (SD)	14.66 (2.40)	14.22 (2.67)	2.08 (0.038)
Sex, *n* (%)	125 (54.35%)	169 (44.47%)	5.21 (0.023)
Hypertension, *n* (%)	81 (35.22%)	284 (74.74%)	91.47 (0.000)
Diabetes, *n* (%)	12 (5.22%)	62 (16.32%)	15.53 (0.000)
BMI, kg/m^2^ (SD)	27.55 (4.75)	26.46 (4.79)	2.76 (0.006)
Depression, score (SD)	2.30 (2.40)	1.60 (1.77)	3.85 (0.000)
IL1β, pg/mL (SD)	0.05 (0.09)	0.04 (0.04)	2.25 (0.026)
IL4, pg/mL (SD)	0.28 (0.25)	0.22 (0.18)	3.06 (0.002)
IL6, pg/mL (SD)	0.69 (2.14)	0.67 (0.80)	0.08 (0.933)
IL8, pg/mL (SD)	2.65 (2.26)	2.90 (2.52)	−1.24 (0.215)
IL10, pg/mL (SD)	0.71 (0.59)	0.59 (0.53)	2.37 (0.018)
NO, pg/mL (SD)	14.14 (7.98)	22.26 (13.13)	−9.50 (0.000)
TAC, pg/mL (SD)	73.24 (18.92)	64.24 (10.48)	6.63 (0.000)
NEO, pg/mL (SD)	7.99 (3.75)	3.01 (2.72)	17.56 (0.000)
MDA, pg/mL (SD)	84.24 (50.98)	85.69 (36.51)	−0.38 (0.706)
GUA, pg/mL (SD)	13.43 (5.59)	12.89 (4.49)	1.26 (0.210)
TNFα, pg/mL (SD)	2.32 (0.95)	2.41 (1.34)	−0.92 (0.357)
TNFR1, pg/mL (SD)	1.05 (0.24)	1.31 (0.49)	−8.78 (0.000)
TNFR2, pg/mL (SD)	1.13 (0.32)	1.43 (0.54)	−8.85 (0.000)
Total tau, pg/mL (SD)	213.41 (901.67)	192.92 (751.16)	0.29 (0.774)
Aβ 38, pg/mL (SD)	206.50 (1142.50)	138.50 (711.93)	0.78 (0.433)
Aβ 40, pg/mL (SD)	179.29 (308.42)	171.78 (164.32)	0.34 (0.733)
Aβ 42, pg/mL (SD)	25.98 (126.73)	17.18 (78.55)	0.95 (0.343)
Aβ 40/42 (SD)	15.13 (5.35)	17.46 (11.40)	−3.41 (0.001)

**Table 2 biomedicines-10-01240-t002:** Associations between principal components of oxidative stress and inflammation and AD biomarkers in the 40s.

	Dependent Variable:									
	Tau		Aβ38 (log)		Aβ40 (log)		Aβ42 (log)		Aβ40/42	
PC1	0.025	0.051	0.084 *	0.101 **	0.051 ***	0.059 ***	0.056 **	0.063 **	−0.005	−0.004
	*p* = 0.663	*p* = 0.399	*p* = 0.085	*p* = 0.050	*p* = 0.002	*p* = 0.0004	*p* = 0.026	*p* = 0.017	*p* = 0.701	*p* = 0.764
PC2	0.129	0.135	0.039	0.058	0.01	0.022	0.017	0.032	−0.007	−0.009
	*p* = 0.104	*p* = 0.101	*p* = 0.543	*p* = 0.385	*p* = 0.650	*p* = 0.313	*p* = 0.615	*p* = 0.368	*p* = 0.663	*p* = 0.595
PC3	0.03	0.018	0.198 **	0.193 **	0.104 ***	0.102 ***	0.136 ***	0.134 ***	−0.032	−0.033
	*p* = 0.767	*p* = 0.861	*p* = 0.019	*p* = 0.024	*p* = 0.0002	*p* = 0.0003	*p* = 0.003	*p* = 0.003	*p* = 0.147	*p* = 0.135
PC4	0.422 ***	0.445 ***	0.249 ***	0.235 ***	0.109 ***	0.110 ***	0.163 ***	0.168 ***	−0.054 **	−0.058 **
	*p* = 0.0001	*p* = 0.00005	*p* = 0.005	*p* = 0.010	*p* = 0.0002	*p* = 0.0002	*p* = 0.0004	*p* = 0.0004	*p* = 0.018	*p* = 0.013
Constant	4.825	5.8	8.421 **	9.627 **	5.978 ***	6.685 ***	3.712 *	4.584 **	2.266 **	2.101 **
	*p* = 0.286	*p* = 0.219	*p* = 0.025	*p* = 0.015	*p* = 0.00001	*p* = 0.00000	*p* = 0.055	*p* = 0.024	*p* = 0.019	*p* = 0.038
Observations	226	226	215	215	230	230	230	230	230	230
Log Likelihood	−429.823	−425.748	−361.794	−360.920	−137.188	−134.408	−242.856	−240.582	−81.986	−80.161
Akaike Inf. Crit.	875.647	877.497	739.588	747.84	290.375	294.817	501.711	507.163	179.971	186.322

Note: * *p* < 0.1; ** *p* < 0.05; *** *p* < 0.01.

**Table 3 biomedicines-10-01240-t003:** Associations between principal components of oxidative stress and inflammation and AD biomarkers in the 60s.

	Dependent Variable:									
	Tau		Aβ38 (log)		Aβ40 (log)		Aβ42 (log)		Aβ40/42	
PC1	−0.127 ***	−0.142 ***	−0.084 **	−0.099 ***	−0.060 ***	−0.065 ***	−0.056 ***	−0.060 ***	−0.005	−0.005
	*p* = 0.008	*p* = 0.005	*p* = 0.011	*p* = 0.004	*p* = 0.00001	*p* = 0.00001	*p* = 0.004	*p* = 0.003	*p* = 0.676	*p* = 0.679
PC2	−0.090	−0.081	0.038	0.04	−0.027	−0.028	−0.027	−0.029	−0.0005	0.002
	*p* = 0.173	*p* = 0.227	*p* = 0.405	*p* = 0.392	*p* = 0.122	*p* = 0.123	*p* = 0.304	*p* = 0.268	*p* = 0.977	*p* = 0.912
PC3	−0.068	−0.076	−0.081	−0.082	−0.001	−0.001	−0.043	−0.043	0.042 **	0.042 **
	*p* = 0.375	*p* = 0.322	*p* = 0.131	*p* = 0.127	*p* = 0.945	*p* = 0.943	*p* = 0.154	*p* = 0.158	*p* = 0.023	*p* = 0.024
PC4	−0.316 ***	−0.291 ***	−0.151 ***	−0.139 **	−0.035	−0.032	−0.061 *	−0.060 *	0.026	0.029
	*p* = 0.0002	*p* = 0.001	*p* = 0.010	*p* = 0.018	*p* = 0.113	*p* = 0.155	*p* = 0.061	*p* = 0.069	*p* = 0.181	*p* = 0.151
Constant	5.648	5.791	6.440 **	7.137 **	4.785 ***	5.219 ***	4.041 **	4.840 ***	0.744	0.379
	*p* = 0.220	*p* = 0.220	*p* = 0.044	*p* = 0.032	*p* = 0.0002	*p* = 0.00005	*p* = 0.027	*p* = 0.010	*p* = 0.496	*p* = 0.736
Observations	375	375	353	353	380	380	380	380	380	380
Log Likelihood	−710.108	−706.354	−531.172	−528.038	−219.326	−216.888	−368.880	−366.495	−176.088	−174.311
Akaike Inf. Crit.	1436.215	1438.708	1078.345	1082.075	454.651	459.777	753.76	758.991	368.175	374.622

Note: * *p* < 0.1; ** *p* < 0.05; *** *p* < 0.01.

## Data Availability

The data presented in this study are available on request from the corresponding author. The data are not publicly available due to ethical and consent restrictions.

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
