# Peer review of "Systemic Inflammation Predicts Alzheimer Pathology in Community Samples without Dementia"

_biomedicines, 2022, doi:10.3390/biomedicines10061240_

Round 1

Reviewer 1 Report

 Biomarkers that could predict Alz disease are a hot topic nowadays. Unfortunately, many factors are still unknown, and it seems to be multifactorial. Homogeneity of the individuals that gave the samples is another obstacle. Maybe, a prospective study with comparisons between the persons in two-time points of their life (e.g. 40s and 60s) could be worthy, although it sounds difficult. 

Author Response

Reviewer 1

Biomarkers that could predict Alz disease are a hot topic nowadays. Unfortunately, many factors are still unknown, and it seems to be multifactorial. Homogeneity of the individuals that gave the samples is another obstacle. Maybe, a prospective study with comparisons between the persons in two-time points of their life (e.g. 40s and 60s) could be worthy, although it sounds difficult.

Response: We agree with this reviewer that longitudinal data would be very useful to investigate change in inflammatory/AD biomarkers. Unfortunately, currently these biomarkers are only available at one timepoint but pending future funding we would certainly like to acquire repeat measurements and conduct the suggested investigations. We have also added this point to the “Implications and future directions” section of the discussion.

Reviewer 2 Report

The paper by Cherbuin and colleagues, entitled ‘Systemic inflammation predicts Alzheimer pathology in com-2 munity samples without dementia’, analyzed the current status of knowledge of the role of neuroinflammation and oxidative stress in Alzheimer’s disease (AD). For this purpose, the associations between specific plasma markers of inflammation, oxidative stress and AD biomarkers (i.e., tau and Amyloid- 38, 40, and 42 levels) were investigated in middle-aged and older-aged patients with dementia. Results showed that in middle-age, a component reflecting variation in OS was most strongly associated with tau and to a lesser extent amyloid- levels, while in older-age, another component reflecting heightened inflammation independent of oxidative stress, was associated with all AD biomarkers. Authors concluded by stating that in middle and older age inflammation and oxidative stress states are associated with plasma AD biomarkers.

The main strength of this paper is that it addresses an interesting and timely question providing a clear answer to the possible role of neuroinflammation and oxidative stress in AD. In general, I think the idea of this article is really interesting and the authors’ fascinating observations on this timely topic may be of interest to the readers of Biomedicines. However, some comments, as well as some crucial evidence that should be included to support the authors’ argumentation, needed to be addressed to improve the quality of the article, its adequacy, and its readability prior to the publication in the present form. My overall judgment is to publish this article after the authors have carefully considered my suggestions below, in particular reshaping parts of the Introduction, Results, and Discussion sections by adding more evidence.

Please consider the following comments:

  • In general, I recommend authors to use more evidence to back their claims, especially in the Introduction of the article, which I believe is currently very lacking. Thus, I recommend the authors to attempt to deepen the subject of their manuscript, as the bibliography is too concise: nonetheless, in my opinion, less than 50 articles for a research paper are really insufficient. Indeed, currently authors cite only 43 papers, and they are dramatically few. Therefore, I suggest the authors to focus their efforts on researching relevant literature: I believe that adding more studies will help to provide better and more accurate background to this study. In this review, I will try to help the authors by suggesting relevant literature that suit their manuscript.
  • Introduction: The ‘Introduction’ section is well-written and nicely presented, with a good balance of descriptive text about molecular and cell biological aspects of AD, and interpretative illustration of the inflammatory process present in most AD patients. Nevertheless, I think that more information about pathophysiology and neurological changes of Alzheimer’s disease would provide a better background here. Thus, I suggest the authors to make such effort to provide a brief overview of the pertinent published literature that offer a perspective on structural and functional correlates of age-associated cognitive changes that might indicate neurodegeneration and lead to dementia, because as it stands, this information is not highlighted in the text. In this regard, I believe that the statement ‘…the extent to which inflammation is attributable to disease progression in the central nervous system (CNS) in the pre- and clinical stages of dementia or emerges in the periphery and is a long-term risk factor for the development of the pathological hallmarks of dementia is an unresolved question’ needs some necessary citations. In particular, according with this sentence, I would recommend citing a recent review that examined pathophysiological basis and biomarkers of AD pathology and investigated molecular signs of neuroinflammation in neurodegenerative diseases, in particular Alzheimer’s disease (https://doi.org/10.3390/ijms21072431). I also recommend a relevant study in which authors investigated age-related impairments in the ability to process contextual information and in the regulation of responses to threat, addressing that structural and physiological alterations in the prefrontal cortex and medial temporal lobe determine cognitive changes in advanced aging, that can eventually cause patterns of cognitive dysfunctions observed in patients with AD/MCI (https://doi.org/10.1038/s41598-018-31000-9). I firmly believe that these improvements will help to provide a more coherent and defined background.
  • Introduction: In according with the previous point raised, when authors stated that ‘there is accumulating evidence indicating that systemic inflammation also contributes to, or at least co-occurs with AD pathophysiology’, I would suggest adding some studies that might address how neuroinflammation correlates spatially with tau aggregation consistently observed in the frontal and/or parietal lobes, causing alterations of frontal lobe that impact memory and error-driven learning in individuals who have an high risk of dementia, may improve the theoretical background of the present article and its argumentation: evidence from an electrophysiological study suggested that medio-frontal ERP signals of prediction error tracks the timing of salient events, and highlighted how alterations in medial prefrontal cortex could impact on the patients’ capacity of signal errors in the prediction of outcomes (https://doi.org/10.1162/jocn_a_01074). Importantly, a recent review on vmPFC subregional contributions addressed the role of vmPFC in processing safety-threat information and their relative value, and how this region is fundamental for the evaluation and representation of stimulus-outcome’s value needed to produce sustained physiological responses (https://doi.org/10.1038/s41380-021-01326-4). Finally, authors can also see studies that have focused on this topic (https://doi.org/10.1162/NECO_a_00779; https://doi.org/10.1038/s41386-021-01101-7).
  • Identification of pro-inflammatory states: I suggest rewriting/reorganizing this paragraph for clarity. In my opinion, authors should present information about principal components of the observed cytokine response in a more clear and structured way, such as using a table or an explanatory figure, because as it stands, this information is not available.
  • Results: Please provide full statistical details to ensure in-depth understanding and replicability of the findings.
  • Discussion: In this section, authors thoroughly described the results and their argumentation and captured the state of the art well; however, I would have liked to see some views on a way forward. Hence, I ask them to include some thoughtful as well as in-depth considerations, making an effort, trying to explain the theoretical as well as the translational application of their research.
  • I believe that the ‘Conclusions’ paragraph would benefit from some thoughtful as well as in-depth considerations by the authors, because as it stands, it is very descriptive but not enough theoretical. Authors should make an effort, trying to explain the theoretical implication as well as the translational application of their research, to adequately convey what they believe is the take-home message of their study, and therefore discussing theoretical and methodological avenues in need of refinement, suggesting a path forward in AD treatment. In this regard, important recent evidence suggests that the application of new methods in Alzheimer’s treatment, such as the Non-invasive brain stimulation techniques (NIBS), have shown promising results in humans (https://doi.org/10.1097/WCO.0000000000000669). Importantly, I recommend citing recent evidence that revealed that the application of NIBS induces long-lasting effects, noninvasively modulating the cortical excitability, and modulates a variety of cognitive functions: for example, a recent review acknowledged the implementation of NIBS to modulate in general fear memories (https://doi.org/10.1016/j.neubiorev.2021.04.036), while another paper highlighted the implementation of NIBS in patients with anxiety disorders that respond poorly to treatment (https://doi.org/10.1016/j.jad.2021.02.076). Finally, I also suggest a review on the efficacy of NIBS (https://doi.org/10.3389/fpsyt.2018.00201).
  • Supplementary Materials: As a reviewer of this manuscript, I would have liked to check Pearson bivariate correlations provided in Figure S1, but the dataset is not available. Please provide a working website.
  • Regarding the Tables: please provide an explanatory caption for each table within the text.
  • The reference list is incorrect: authors should check the Journal’s guidelines again and provide the abbreviated journal name in italics, the year of publication in bold, the volume number in italics.
  • The number of the pages on the manuscript is incorrect and requires to be fixed right after Table 2.

Overall, the manuscript contains 1 figure, 2 tables, and 43 references. In my opinion, the number of references it is too low for a research article, and this prevents the possibility of publishing it in this form. References should be more than 50 for original research articles.

However, the manuscript carries important value presenting an association between neuroinflammation and oxidative stress (OS) and plasma AD biomarkers and could be a valid paper for the journal.

I hope that, after these careful revisions, the manuscript can meet the Journal’s high standards for publication. I am available for a new round of revision of this article.

Best regards,

Reviewer

Author Response

Reviewer 2

The paper by Cherbuin and colleagues, entitled ‘Systemic inflammation predicts Alzheimer pathology in com-2 munity samples without dementia’, analyzed the current status of knowledge of the role of neuroinflammation and oxidative stress in Alzheimer’s disease (AD). For this purpose, the associations between specific plasma markers of inflammation, oxidative stress and AD biomarkers (i.e., tau and Amyloid- 38, 40, and 42 levels) were investigated in middle-aged and older-aged patients with dementia. Results showed that in middle-age, a component reflecting variation in OS was most strongly associated with tau and to a lesser extent amyloid- levels, while in older-age, another component reflecting heightened inflammation independent of oxidative stress, was associated with all AD biomarkers. Authors concluded by stating that in middle and older age inflammation and oxidative stress states are associated with plasma AD biomarkers.

The main strength of this paper is that it addresses an interesting and timely question providing a clear answer to the possible role of neuroinflammation and oxidative stress in AD. In general, I think the idea of this article is really interesting and the authors’ fascinating observations on this timely topic may be of interest to the readers of Biomedicines. However, some comments, as well as some crucial evidence that should be included to support the authors’ argumentation, needed to be addressed to improve the quality of the article, its adequacy, and its readability prior to the publication in the present form. My overall judgment is to publish this article after the authors have carefully considered my suggestions below, in particular reshaping parts of the Introduction, Results, and Discussion sections by adding more evidence.

Response: We are grateful to this reviewer for their suggestions aimed at improving our paper.

Please consider the following comments:

Point 1:  In general, I recommend authors to use more evidence to back their claims, especially in the Introduction of the article, which I believe is currently very lacking. Thus, I recommend the authors to attempt to deepen the subject of their manuscript, as the bibliography is too concise: nonetheless, in my opinion, less than 50 articles for a research paper are really insufficient. Indeed, currently authors cite only 43 papers, and they are dramatically few. Therefore, I suggest the authors to focus their efforts on researching relevant literature: I believe that adding more studies will help to provide better and more accurate background to this study. In this review, I will try to help the authors by suggesting relevant literature that suit their manuscript.

Response: We agree that it is important to provide concrete evidence to support claims and discussions. In recent times, our experience is that journals ask to have fewer references and shorter articles, not more and longer. However, we are very happy to extend our bibliography and supporting evidence. As suggested, we have added many references to the manuscript which now total 76, see also related Points 2, 3, 4, and 7.

Point 2: Introduction: The ‘Introduction’ section is well-written and nicely presented, with a good balance of descriptive text about molecular and cell biological aspects of AD, and interpretative illustration of the inflammatory process present in most AD patients. Nevertheless, I think that more information about pathophysiology and neurological changes of Alzheimer’s disease would provide a better background here. Thus, I suggest the authors to make such effort to provide a brief overview of the pertinent published literature that offer a perspective on structural and functional correlates of age-associated cognitive changes that might indicate neurodegeneration and lead to dementia, because as it stands, this information is not highlighted in the text.

Response: As suggested we have provided much more detailed background to the pathophysiology of the disease, its association with changes in brain structure and cognitive function, and their link and predictive value of future cognitive decline and dementia (see introduction, pages 1-3).

Point 3: In this regard, I believe that the statement ‘…the extent to which inflammation is attributable to disease progression in the central nervous system (CNS) in the pre- and clinical stages of dementia or emerges in the periphery and is a long-term risk factor for the development of the pathological hallmarks of dementia is an unresolved question’ needs some necessary citations. In particular, according with this sentence, I would recommend citing a recent review that examined pathophysiological basis and biomarkers of AD pathology and investigated molecular signs of neuroinflammation in neurodegenerative diseases, in particular Alzheimer’s disease (https://doi.org/10.3390/ijms21072431). I also recommend a relevant study in which authors investigated age-related impairments in the ability to process contextual information and in the regulation of responses to threat, addressing that structural and physiological alterations in the prefrontal cortex and medial temporal lobe determine cognitive changes in advanced aging, that can eventually cause patterns of cognitive dysfunctions observed in patients with AD/MCI (https://doi.org/10.1038/s41598-018-31000-9). I firmly believe that these improvements will help to provide a more coherent and defined background.

Response: Thank you for these suggestions. We have now referenced the above review. However, we felt that the study on processing of contextual information in response to existential threat was not sufficiently relevant to the focus present study to warrant inclusion.

Point 4: Introduction: In according with the previous point raised, when authors stated that ‘there is accumulating evidence indicating that systemic inflammation also contributes to, or at least co-occurs with AD pathophysiology’, I would suggest adding some studies that might address how neuroinflammation correlates spatially with tau aggregation consistently observed in the frontal and/or parietal lobes, causing alterations of frontal lobe that impact memory and error-driven learning in individuals who have an high risk of dementia, may improve the theoretical background of the present article and its argumentation: evidence from an electrophysiological study suggested that medio-frontal ERP signals of prediction error tracks the timing of salient events, and highlighted how alterations in medial prefrontal cortex could impact on the patients’ capacity of signal errors in the prediction of outcomes (https://doi.org/10.1162/jocn_a_01074). Importantly, a recent review on vmPFC subregional contributions addressed the role of vmPFC in processing safety-threat information and their relative value, and how this region is fundamental for the evaluation and representation of stimulus-outcome’s value needed to produce sustained physiological responses (https://doi.org/10.1038/s41380-021-01326-4). Finally, authors can also see studies that have focused on this topic (https://doi.org/10.1162/NECO_a_00779; https://doi.org/10.1038/s41386-021-01101-7).

Response:  As suggested, we have added more evidence reporting on the spatial association between inflammation and tau aggregation and abeta deposition:

[page 3-4, lines 106-124]

“The precise temporal and spatial relationship between these processes is not well understood but emerging evidence suggests that inflammation may precede Aβ deposition and may be caused or exacerbated by increased tau phosphoralisation. Indeed, a recent study[27] investigating the associations between central inflammation with 11C-PK11195 Positron Emission Tomography (PET), Aβ (11C-PiB PET), and tau (18F-Flortaucipir PET) deposition in MCI over a 2-year period indicates that neuroin-flammation may occur in the early AD pre-clinical stages, and is followed by a spatially correlated increase in Aβ in the frontal, temporal, and occipital lobes. Interestingly, these associations do seem to persist when controlling for tau load. In contrast, central inflammation was found to be spatially correlated with tau deposition across the whole follow-up, first in the frontal and temporal lobes, and later in the occipital lobe. However, unlike for Aβ, associations between central inflammation and tau did not change after controlling for Aβ. This appears to suggest that inflammation may be more strongly associated with tau pathology than Aβ. Evidence from bio-chemistry, in-vitro, animal and other human studies also seem to bring support to this hypothesis[28], and consistent evidence has also been demonstrated in fronto-temporal dementia[16].”

And

[page 4, lines 126-131]

“In addition, there is accumulating evidence indicating that systemic inflammation also contributes to, or at least co-occurs with AD pathophysiology[29,30]. In animal models, systemic inflammation has been shown to be associated with glial activation, tau hyperphosphoralisation in the frontal cortex and the hippocampus, and Aβ deposition in the hippocampus[31,32]. While human studies have reported associations between markers of systemic inflammation and PET markers of Aβ and tau deposition[33].”

However, relating pathological hallmarks with detailed cognitive sub-domains and their relationship with ERP is beyond the scope of this discussion as it would require systematically reviewing all major domains as well as the many modalities to assess them, of which there are many.

Point 5: Identification of pro-inflammatory states: I suggest rewriting/reorganizing this paragraph for clarity. In my opinion, authors should present information about principal components of the observed cytokine response in a more clear and structured way, such as using a table or an explanatory figure, because as it stands, this information is not available.

Response: Apologies. This information was already available in the supplementary methods of the previous version but we had forgotten to refer to it in this section of the manuscript. We felt that it was more appropriate to report this information in supplementary material because it is quite lengthy and technical and would likely make the article harder to read for the typical reader if it was included in the main body of the manuscript. We have now clearly referenced this material.

Point 6: Results: Please provide full statistical details to ensure in-depth understanding and replicability of the findings.

Response: It is not clear to us what additional statistical results would like to see. We have provided the statistics of all our results in two tables within the main manuscript, and in several more tables and figures in the supplementary material. We would be happy to clarify further if needed but we could not identify missing statistics.

Point 7: Discussion: In this section, authors thoroughly described the results and their argumentation and captured the state of the art well; however, I would have liked to see some views on a way forward. Hence, I ask them to include some thoughtful as well as in-depth considerations, making an effort, trying to explain the theoretical as well as the translational application of their research. I believe that the ‘Conclusions’ paragraph would benefit from some thoughtful as well as in-depth considerations by the authors, because as it stands, it is very descriptive but not enough theoretical. Authors should make an effort, trying to explain the theoretical implication as well as the translational application of their research, to adequately convey what they believe is the take-home message of their study, and therefore discussing theoretical and methodological avenues in need of refinement, suggesting a path forward in AD treatment. In this regard, important recent evidence suggests that the application of new methods in Alzheimer’s treatment, such as the Non-invasive brain stimulation techniques (NIBS), have shown promising results in humans (https://doi.org/10.1097/WCO.0000000000000669). Importantly, I recommend citing recent evidence that revealed that the application of NIBS induces long-lasting effects, noninvasively modulating the cortical excitability, and modulates a variety of cognitive functions: for example, a recent review acknowledged the implementation of NIBS to modulate in general fear memories (https://doi.org/10.1016/j.neubiorev.2021.04.036), while another paper highlighted the implementation of NIBS in patients with anxiety disorders that respond poorly to treatment (https://doi.org/10.1016/j.jad.2021.02.076). Finally, I also suggest a review on the efficacy of NIBS (https://doi.org/10.3389/fpsyt.2018.00201).

Response: As suggested we have added a discussion of the theoretical implications and need for further research by adding an “Implications and future directions” section as follows (page 14, line 410):

“Implications and Future Directions

A direct implication of the present findings is that, where possible, assessment of inflammation through multiple cytokines and chemokines is preferable to the use of single markers, or to the separate analysis of individual biomarkers. These results also raise important conceptual questions. Since cross-sectional analyses were conducted to assess associations between inflammatory states and AD biomarkers, it is not clear whether systemic inflammation is a consequence of the developing AD pathology, or whether an increase in peripheral inflammation led to, or promoted the development of AD pathology, or indeed, whether these processes are directly or causally related at all. Although there is theoretical support for all these hypotheses and they do not need to be mutually exclusive, based on current evidence it seems more likely that inflammation originating in the periphery has an effect on central pathological processes rather than the reverse. This is because there is robust evidence indicating that conditions associated with heightened peripheral inflammation, such as sepsis, osteo-arthritis, and systemic lupus erythematosus, are associated with increased neuro-degeneration, cognitive decline and dementia[73-76]. Moreover, controlled experiments in AD mice models have showed that low-grade systemic inflammation is associated with greater glial activation, decreased Ab clearance, and greater Ab deposition[38,40]. In contrast, there is little, if any, evidence demonstrating that increased AD pathology, may lead to increased systemic inflammation.

Nonetheless, there is a need for human studies investigating longitudinally the associations between systemic inflammation, increased AD pathology in the brain, and plasma AD biomarkers over longer timeframes and with several repeated assessments such that directional associations can be tested. Another research gap that requires more attention is the specific role OS plays in AD pathophysiology. OS is of-ten conflated with inflammation and the present findings suggest that more routinely identifying the discrete effects attributable to each of these processes may prompt new insights into the timing and the origin of AD pathological development, and thus provide a better understanding of the optimal preventative window and of the more promising risk reduction strategies and targets.”

Point 8: Supplementary Materials: As a reviewer of this manuscript, I would have liked to check Pearson bivariate correlations provided in Figure S1, but the dataset is not available. Please provide a working website.

Response: As noted in the data availability section, data from this study are not publicly available. However, we have uploaded the requested data with other manuscript files for your review.

Point 9: Regarding the Tables: please provide an explanatory caption for each table within the text.

Response: We are unclear about what is requested here. Tables 1, 2 & 3 and supplementary Tables S1, S2, and S3 all have a legend and we are unsure what additional information this reviewer would like to see.

Point 10: The reference list is incorrect: authors should check the Journal’s guidelines again and provide the abbreviated journal name in italics, the year of publication in bold, the volume number in italics.

Response: Apologies for this error in formatting. We had used the MDPI endnote style but perhaps it was one suitable for another journal. We have now downloaded the specific style for Biomedicines (https://www.mdpi.com/authors/references) and applied it to our manuscript.

Point 11: The number of the pages on the manuscript is incorrect and requires to be fixed right after Table 2.

Response: We have corrected this numbering issue as suggested. However, we will likely need assistance from the editorial team as some idiosyncrasies of the MDPI template are beyond our expertise.

Point 12: Overall, the manuscript contains 1 figure, 2 tables, and 43 references. In my opinion, the number of references it is too low for a research article, and this prevents the possibility of publishing it in this form. References should be more than 50 for original research articles.

However, the manuscript carries important value presenting an association between neuroinflammation and oxidative stress (OS) and plasma AD biomarkers and could be a valid paper for the journal.

I hope that, after these careful revisions, the manuscript can meet the Journal’s high standards for publication. I am available for a new round of revision of this article.

Response: Thank you for your careful and detailed review which has helped improve our manuscript. We have addressed these issues in detail in the above responses.

Reviewer 3 Report

Neuroinflammation and oxidative stress are implicated in the pathophysiology of Alzheimer’s disease.

Nicolas Cherbuin et al. has investigated Systemic inflammation predicts Alzheimer pathology in community samples without dementia.

 In general the manuscript contain relevant paragraphs that have been discussed. The selection of bibliography is appropriate to the content of the manuscript. In the conclusion, the authors have included short thoughts. Perhaps even too brief reflections.

Finally, regarding methodology, authors refer about statistics thus the readers can make assumptions regarding the quality and the confidence of the results and reasonability of consideration of the authors. Is such a large standard deviation acceptable in Table 1? This raises my question.

This study was conducted clearly and provided some interesting results about the associations between specific plasma markers of inflammation and oxidative stress, but there are a couple of concerns. The manuscript is very enjoyable to read, but after close evaluation of the paper I suggest revision according to the next point:

  1. Indicate the element of novelty.

  1. Please correct some parts of the manuscript.

 “Markers of inflammation: Tumor Necrosis Factor alpha and TNF receptors (TNF , TNF-R1, TNF-R2), and interleukins (IL1 , IL4, IL6, IL8, IL10).

“Serum/plasma samples were collected after a fast of at least 8 hours, stored at -80C aliquoted in 1ml vials. Immediately prior to analysis samples were thawed and were all processed with the same pipeline at the same time at the ANU Phenomics Facility (see supplementary material for details).”

“Age, total years of education, diabetes mellitus, depression symptomatology (Goldberg depression)[26], and smoking (ever) were assessed by self-report. Body mass index (BMI) was computed with the formula weight (kg)/height x height (m2) based on self  report of weight and height.”

I am not a native English speaker, so I don't know if the manuscript would benefit from careful and profound proof-reading and correction of language and style.

Author Response

Reviewer 3

In general the manuscript contain relevant paragraphs that have been discussed. The selection of bibliography is appropriate to the content of the manuscript. In the conclusion, the authors have included short thoughts. Perhaps even too brief reflections.

Finally, regarding methodology, authors refer about statistics thus the readers can make assumptions regarding the quality and the confidence of the results and reasonability of consideration of the authors. Is such a large standard deviation acceptable in Table 1? This raises my question.

This study was conducted clearly and provided some interesting results about the associations between specific plasma markers of inflammation and oxidative stress, but there are a couple of concerns. The manuscript is very enjoyable to read, but after close evaluation of the paper I suggest revision according to the next point:

Response:

Thank you for your thoughtful review and your suggested improvements to our manuscript.

Point 1: Indicate the element of novelty.

Response: As suggested we have indicated the novel nature of this study as follows (page 11, lines 295-302):

“This study has applied a very novel approach to investigate how systemic inflammation relates to biomarkers of two AD hallmarks. By considering inflammatory states, identified through PCA and assessed through the combined contribution of several markers of inflammation and OS (i.e. not single cytokines individually), it ensured that that the complex inter-relationship between cytokines and chemokines involved in varying inflammatory responses would be captured in a more holistic way. Another novel aspect of this research is that these relationships were investigated in community-living individuals without dementia.”

Point 2: Please correct some parts of the manuscript.

 “Markers of inflammation: Tumor Necrosis Factor alpha and TNF receptors (TNF , TNF-R1, TNF-R2), and interleukins (IL1 , IL4, IL6, IL8, IL10).”

“Serum/plasma samples were collected after a fast of at least 8 hours, stored at -80C aliquoted in 1ml vials. Immediately prior to analysis samples were thawed and were all processed with the same pipeline at the same time at the ANU Phenomics Facility (see supplementary material for details).”

“Age, total years of education, diabetes mellitus, depression symptomatology (Goldberg depression)[26], and smoking (ever) were assessed by self-report. Body mass index (BMI) was computed with the formula weight (kg)/height x height (m2) based on self  report of weight and height.”

Response: As suggested we have corrected the noted sections as follows:

“Markers of inflammation: Tumour Necrosis Factor alpha (TNF-a) and TNF receptors (TNF-R1, TNF-R2), and interleukins (IL1b, IL4, IL6, IL8, IL10).”

“Serum/plasma samples were collected after a fast of at least 8 hours, stored at -80C, and aliquoted in 1ml vials. Immediately prior to analysis, samples were thawed and were all processed with the same pipeline at the same time at the ANU Phenomics Facility (see supplementary material for details).”

“Age, total years of education, diabetes mellitus, depression symptomatology (Goldberg depression)[26], and smoking (ever) were assessed by self-report. Body mass index (BMI) was computed with the formula weight (kg)/height x height (m2) based on self-report of weight and height.”

Point 3: I am not a native English speaker, so I don't know if the manuscript would benefit from careful and profound proof-reading and correction of language and style.

Response: Thank you for your candid acknowledgement. Several authors on this manuscript are native English speakers and have carefully reviewed the manuscript. We are confident that language and style are of very high quality.